

# Acidification counteracts negative effects of warming on diatom silicification

Alexandra Coello-Camba[1], Susana Agustí[1,2]

[1] Red Sea Research Center, King Abdullah University of Science and Technology (KAUST), Thuwal, 23955-6900, Kingdom of Saudi Arabia.
[2] Department of Arctic and Marine Biology, Faculty of Bioscience, Fishery and Economy, University of Tromsø, Tromsø, 9037, Norway.

*Correspondence to*: A. Coello-Camba (Alexandra.camba@kaust.edu.sa)

**Abstract.** Diatoms are a significant group contributing up to 40% of annual primary production in the oceans. They have a special siliceous cell wall that, acting as a ballast, plays a key role in the sequestration of global carbon and silica. Diatoms dominate primary production in the Arctic Ocean, where global climate change is causing increases in water temperature and in the partial pressure of $CO_2$ ($pCO_2$). Here we show that as water temperature increases diatoms become stressed, grow to smaller
sizes, and decrease their silicification rates. But at higher $pCO_2$, as the pH of seawater decreases, silica incorporation rates are increased. In a future warmer Arctic ocean diatoms may have a competitive advantage under increased ocean acidification, as increased $pCO_2$ counteracts the adverse effects of increasing temperature on silicification and buffers its consequences in the biogeochemical cycles of carbon and silica.

**Keywords**: Diatoms, Silica, Interaction, Temperature, $pCO_2$

## 1. Introduction

The Arctic Ocean is experiencing a strong warming process (ACIA, 2004), with some scenarios predicting mean annual temperatures in central Arctic to increase more than 5 ºC by the end of this
century (ACIA, 2004). Besides this, the concentration of atmospheric $CO_2$ has also increased from about 280 ppm in the pre-industrial era to close to 400 ppm at present (Forster and Ramaswamy, 2007), being largely responsible for the documented decrease in seawater pH. Future scenarios anticipate $pCO_2$ values between 700 - 1000 ppm by the year 2100 (Caldeira and Wickett, 2003). But, despite the particular vulnerability of polar ecosystems, studies in the Arctic Ocean are still rather scarce
(Wassmann et al., 2010).
Diatoms contribute up to 40% of annual primary production in the oceans (Mann, 1999), and are central to the Arctic spring bloom (Gosselin et al., 1997; Hodal et al., 2012), when the bulk of annual primary production takes place (Piwosz et al., 2009; Vaquer-Sunyer et al., 2013). Diatoms are cells comprised by two silica ($SiO_2$) valves with important protective functions (Hamm et al., 2003; Raven and Waite,
2004) and a crucial influence on their sinking capacity (Smetacek 1985) due to their higher density.



Diatoms are the main producers of biogenic silica in the photic layer as a consequence of the buildup of their valves (Tréguer and De la Rocha, 2013). Part of this silica is directly recycled in the surface and part is exported, slowly dissolving as it sinks and accumulates at the deep ocean (Tréguer and De la Rocha, 2013). In total, diatoms precipitate in the world's oceans approximately 240 Tmol of silica per
year (Tréguer and De la Rocha, 2013). The export of particulate organic carbon from the euphotic zone is highly correlated with this flux of particulate biogenic silicate (Falkowsky et al., 2003). This way diatoms contribute considerably to the transfer of silicic acid from the hydrosphere to the biosphere (Tréguer et al., 1995) and to the export of organic material from the surface to the sea floor (Smetacek, 1985; Turner, 2002).
Several environmental factors can induce changes in diatom valve architecture and silica metabolism. Among these factors, higher oceanic temperatures have been shown to have a negative effect on diatom silicification (Kamatani, 1982; Montagnes and Franklin, 2001), and a relationship has also been identified between changes in pH and changes in silicification-related features (Lewin, 1961; Hervé et al., 2012). But, to date, there is insufficient evidence defining the relationship between silicification and
the partial pressure of $CO_2$ ($pCO_2$) (Milligan et al., 2004).
Our analysis focuses on how increased temperatures and $pCO_2$ affect silica content and its incorporation rates into Arctic diatoms, with implications on the export rate of diatom-derived organic matter to the deep sea. Increased temperature and $pCO_2$ have been observed to have interactive effects on several diatom features (i. e. Torstensson et al., 2013), and here we also searched for possible interactions
between both stressors. We perform three *in situ* experiments with natural communities of Arctic diatoms (without silica limitation) exposed to temperature gradients of 1-10.5 ºC and different levels of $pCO_2$ (Tables 1 and 2). We analyze the effects of temperature and $pCO_2$ on cell and valve dimensions (Fig. 1A) and silicification, testing for possible interactions when both stressors act jointly.

### 2. Methods

The three experiments were run at the University Centre in Svalbard (UNIS) in Longyearbyen during the summers of 2009 and 2010. They involved three communities of Arctic diatoms (Coello-Camba et al., 2014 and 2015) (Table 1).

### 2.1 Experimental set up

Seven temperature treatments were set for the 2009 experiments and three temperature treatments in 2010 (Table 2); in this last experiment temperatures were combined with two $pCO_2$ targets, 380 ppm ("present $pCO_2$") and 1000 ppm ("increased $pCO_2$").
All plastic and glassware used for the incubations was previously cleaned with HCl and thoroughly rinsed with seawater. Seawater was mixed in 280 L containers, screened through a 100 μm
mesh to remove large grazers, and transferred to 20 L Nalgene $^{TM}$ polycarbonate bottles. These bottles were submersed in 280 L tanks connected to temperature control units (PolyScience 9600 series, precision 0.1 ºC), with impelling and expelling pumps. The temperature stability was monitored along the experiment with data loggers submersed in each tank. The target $pCO_2$ level was achieved by fitting



each experimental 20 L bottle with a bubbling system connected to $CO_2$ bottles and air mixture bottles. The gas mixture was provided by mass flow controllers (model GFC17, Aalborg Instruments and Controls, Inc.), setting a flow rate of 0-10 L min$^{-1}$ for air mixture and 0-10 mL min$^{-1}$ for $CO_2$, (in the case of the air mixture, an aquarium pump forced the gas to pass through soda lime to remove all $CO_2$

before reaching the mass flow controller). Once the proper flow was set, both gasses were mixed, introduced with plastic tubes in each 20 L bottle and bubbled from the bottom of the bottles.
The experimental bottles were illuminated with two fluorescent light tubes per tank to provide an appropriate, continuous light environment (200 µmol photons PAR m$^{-2}$ s$^{-1}$, 24 hours) throughout the experiment.

**2.2 Cell size and valve thickness**

To evaluate the presence of changes in cell size and/or valve thicknesses, approximately 0.5 L were sampled from each treatment at the end of each experiment and then concentrated through filtration to a final volume of 2 mL; these samples were frozen until analysis.
Prior to microscopic observations, and to facilitate the measurement of valve sizes and thicknesses, cells

were cleaned through a digestion process. For this purpose, samples were incubated for 2 h at 18 ºC in $H_2O_2$ 3% (125 µL per mL sample) and then centrifuged for 20 min (2500 rpm). The resulting precipitate was resuspended to 2 mL in filtered seawater. An enzymatic method was then applied to completely break up the organic components of the cells and to clean the valves. These were incubated 15 min at 25 ºC with DNAse (200 µL mL$^{-1}$) and 30 min at 25 ºC with trypsin (200 µL mL$^{-1}$). After digestion,

enzymes were removed by several successive centrifugations (20 min, 2500 rpm), replacing the resulting supernatant with filtered seawater.
Once the valves had been cleaned and emptied and the cell debris had been removed, clean samples were processed using a Zeiss Axioplan 2 Imaging microscope equipped with a 100x objective and a magnifying filter Optovar 1.6x. A Spot Insight B/W camera was used to obtain images from all clean

diatom valves, and sample imaging and measurements were performed using Metamorph software (Meta Imaging Series, version 5.0r1).
We measured valve thicknesses and cell diameters and calculated the corresponding cell biovolumes by approximation to the closest geometric figure.

**2.3 Rates of incorporation of new silica into cell valves**

To estimate possible alterations in the rate at which new silica is incorporated into the valves of diatoms we used the PDMPO (Lysosensor™ Yellow/Blue DND-160, L7545, Molecular Probes®) staining technique during the ATP 2010 experiment. This technique stands out as one of the simplest and more efficient methods for studying the process of silicification in diatoms (Leblanc and Hutchins, 2005; McNair et al., 2015). This compound quickly enters the cells and is specifically incorporated into their

newly synthesized valves, acting as a unique fluorescent biosilicification tracer (Shimizu et al., 2001). Initial, intermediate and final samples were taken to determine the rate at which newly synthesized silica was being incorporated into the valves of diatoms. These samples were stained and processed following the method described by Leblanc and Hutchins (2005): 250 mL of sample was incubated with PDMPO (to a final concentration of 0.125 µM) for 24 h under the corresponding temperature and $pCO_2$





conditions for each treatment. Samples were then filtrated through white polycarbonate filters with a pore size of 1 μm, carefully cleaned with filtered seawater, and kept frozen in vials until analysis. PDMPO was extracted from the filters following a hot NaOH digestion method: filters were placed in 15 mL tubes and 4 mL of NaOH (0.2 N) was added, and samples were incubated at 95 ºC for 1 h. After

digestion, the tubes were transferred to an ice bath and, after cooling, 1 mL of HCl 1N was added to neutralize NaOH. Filters were carefully pushed to the bottom of the tubes with a plastic spatula and centrifuged for approximately 10 min at 300 rpm to remove any accumulation of organic matter or debris that could alter the measurements. A few milliliters of the supernatant were then used to read PDMPO fluorescence directly on the spectrofluorometer (AMINCO-Bowman™ series II) at 350 nm

excitation and 527 nm emission wavelengths.

Based on the methodology described by Shimizu et al. (2001), a calibration line was obtained by measuring the fluorescence of a series of dilutions containing different concentrations of silicic acid, ranging from 0 to 6 μM with a known concentration of PDMPO (0.6 nM). A stock dilution of silicic acid (106 mM) was obtained by passing a sodium silicate solution ($Na_2SiO_3$, 0.5M) through a cation-

exchange resin Dowex® 50W x8, hydrogen form, and then diluted to the final concentrations for calibration. As a result, we obtained the following equation:

$$[BSi] (\mu M) = 2.47 \cdot [PDMPO] (nM) - 0.48, \tag{1}$$

PDMPO incorporation rates were estimated considering the difference in PDMPO concentration measured at the beginning and at the end of the 24-h incubation period.

To test how interactions between both stressors act on the silica incorporation rates of diatoms, we applied the probabilistic Independent Action model described in Payne et al. (2001). This model

allowed us to calculate the predicted effects ($e_{mix}$) of a mixture of known levels of temperature and $pCO_2$ on incorporation rates by using the expression:

$$e_{mix\ (i,j)} = 1 - [1 - E_i][1 - E_j], \tag{2}$$

where $E_i$ is the effect produced by a specific temperature and $E_j$ is the effect produced by a specific $pCO_2$ level when each are applied alone.

We used the incorporation rate observed under control conditions (1.8 ºC and 217.7 ppm) during the experiment as a reference point. The effects of increased temperature and $pCO_2$ were expressed relative to the maximal effect (Payne et al., 2001):


$$E_i = e_i/e_{max}, \tag{3}$$

This is a model of joint effects that assumes additivity and denotes synergy by positive deviations of the observed from the predicted effect ($e_{mix}$). Thus, we considered synergistic effects of increased

temperature and $pCO_2$ when the observed incorporation rates were significantly higher than the predicted incorporation rates. To analyze the significance of the differences between predicted and observed effects, we applied an ANOVA test using the JMP software.



### 2.4 Estimation of sinking rates

We used the equation described in Miklasz and Denny (2010) to predict the sinking speed of a diatom from its cell radius and valve thickness:

$$U = \frac{2g}{9\mu}\left[\rho_{cyt}\frac{(r-t)^3}{r} + \rho_{fr}\frac{(3r^2t - 3rt^2 + t^3)}{r} - \rho_w r^2\right],\tag{4}$$

where U is the sinking speed (m d$^{-1}$), g is the gravitational acceleration, $\mu$ is the dynamic viscosity of seawater at 20 ºC and 33 g L$^{-1}$ salinity (1.07 x 10$^{-3}$ Pa), $\rho_{cyt}$ is the median density of the cytoplasm in marine diatoms (1065 kg m$^{-3}$), r is the cell radius, t is valve thickness, $\rho_{fr}$ is the median cell wall density in diatoms (1800 kg m$^{-3}$), and $\rho_w$ is the density of seawater at 20 ºC and 33 g L$^{-1}$ salinity (1023 kg m$^{-3}$).

### 2.5 Total silicate concentration

To measure the changes in total silicate concentration, each bottle was sampled at the beginning of each experiment and every second day. Samples were kept frozen until their analysis using standard methods (Hansen and Koroleff, 1999) in a Bran Luebe AA3 autoanalyzer.

Initial silicate concentrations for the first and second experiment (ATP 2009) were 0.19 and 0.41 µmol L$^{-1}$, respectively. For the last experiment (ATP 2010), initial silica concentration was higher, 0.61 µmol L$^{-1}$.

### 2.6 Data analysis

The statistical analysis of data was performed using JMP statistical software.

### 3. Results

The incubation temperatures set for each treatment remained very stable along the three experiments performed here, except for one treatment in the fjord community experiment that had to be discarded after a malfunction of the cooling system (Table 2). With regard to the $p$CO$_2$ treatments, pH values varied between 8.2, 7.9 and 7.5 pH units.



### 3.1 Cell size and valve thickness

Valve measurements were performed on the centric diatoms most abundantly observed in our samples after a process of cell cleaning (Fig. 1A); centric diatoms with 21.4 µm in diameter from the 2009 open sea community experiment; centric diatoms with 7.4 µm in diameter from the 2009 fjord
community experiment, and 6.6 µm in diameter from the 2010 experiment. Valve thicknesses (VT) varied between 343 (± 2) nm for the centric diatoms 21.4 µm, 208 (± 1) for the centric diatoms 7.4 µm, and 190 (± 1) for the centric diatoms 6.6 µm in diameter. Other groups as centric diatoms 37.1 µm in diameter, *Nitzschia sp.* and *Chaetoceros spp.* were also present, but because of their low abundance these cells were not considered in our analysis.
One of the most evident results was the significant decrease in diatom cell volumes (CV) (Table 3; Fig. 1B) under increased temperatures ($p < 0.05$), leading to a decrease in cell size of about 1.9 % (± 0.6) for each ºC increase. Increased $p$CO$_2$ showed no significant effects on cell size (Table 3). Diatoms kept a certain constancy of valve thicknesses under increased temperature, showing minor changes only significant for the group of centric diatoms 7.4 µm in diameter ($p < 0.05$) (Table 3; Fig.
1C). The effect of increased $p$CO$_2$ was significant ($p < 0.001$), although very slight (Table 3).

### 3.2 Silica incorporation rates

We observed a decreasing trend in the incorporation rates of BSi to the cell valves with increasing temperature ($p<0.05$) (Fig. 2A). An opposite response was found, with higher incorporation
rates under increased $p$CO$_2$ (Fig. 2B).
   By applying the IA model, we compared the separate effects of increased temperature and $p$CO$_2$ (Fig. 3A), with the additive joint effects predicted by the model (Fig. 3B) and the joint effects actually observed (Fig. 3C). Here, the model showed that additivity would result in a negative effect of both stressors on the incorporation rates under all treatments (Fig. 3B); however, observed results were
different, showing a positive joint effect at 6.7 ºC (Fig. 3C).

### 3.3 Sinking rates

The average sinking rate (U) calculated for centric diatoms 21.4 µm was 2.28 (±0.06) m d$^{-1}$, for centric diatoms 7.4 µm was 0.38 (±0.01) m d$^{-1}$, and for centric diatoms 6.6 µm was 0.32 (±0.01) m d$^{-1}$.
A decreasing trend in the sinking rates was also observed here under increasing temperatures but only statistically significant for the largest diatom (21.4 µm in diameter, $p < 0.05$) (Table 3). The effect of increased $p$CO$_2$ was highly significant ($p < 0.001$), but also very slight (Table 3).





### 4. Discussion

Cell volumes (CV) (Fig. 1A) for the dominant diatom groups in the community showed a significant decrease in diameter with increasing temperature ($p < 0.05$), in agreement with previous observations (Lewin, 1961; Atkinson et al., 2003). Overall cell size decreased about 1.9 % ($\pm$ 0.6) for each 1ºC

increase (Table 3), similar to the percentage observed by Atkinson et al. (2003). The size of diatoms has been shown to typically decrease with successive cell cycles independent of temperature (Edlund and Stoermer, 1997), but recent studies support the idea that increasing temperatures may induce changes in the balance between anabolic and catabolic processes that can alter organism size (Atkinson and Sibly, 1997). Reducing cell size increases the surface:volume ratio to compensate for this imbalance (Atkinson

et al., 2003).

The microscopic measurement of the thickness of silica valves (Fig. 1A) showed an absence of a scaling relationship between cell size and valve thickness (Miklasz and Denny, 2010), with diatoms maintaining a certain constant valve thicknesses despite their decreasing cell size (Table 1; Fig. 1A and

C).

Valve thickness and cell size influence the processes of silica uptake and usage of the cells, and thus control the degree of diatom silicification. To estimate possible changes in the rate at which new silica would be incorporated into the valves, we used the PDMPO (Lysosensor™ Yellow/Blue DND-160,

L7545, Molecular Probes®) staining technique. This compound specifically incorporates into newly synthesized valves, acting as a unique fluorescent biosilicification tracer (Shimizu et al., 2001). Incorporation rates of biogenic silica (BSi) into the cell valves (Fig. 2) were consistent with previous estimations for natural communities with similar initial silicate concentrations (Chisholm et al., 1978). With increasing temperature, we observed a trend for these rates to decrease (Fig. 2A) consistent with a

decrease in silica incorporation rates due to their smaller size under high temperatures (Hildebrand, 2005).
Alternatively, under increased $p\text{CO}_2$, we observed increased rates of BSi incorporation into the cell valves (Fig. 2B). This result was expected because several biological processes that regulate silicon metabolism have been shown to be sensitive to external pH changes (Hervé et al., 2012): low pH

prevents silica wall dissolution (Vrieling et al., 1999) and increases silica incorporation rates (Hervé et al., 2012). Here, by increasing $p\text{CO}_2$ levels we achieved pH values (of 8.2, 7.9 and 7.5 units) that ultimately resulted in increased silica incorporation rates.
Increased $p\text{CO}_2$ mildly affected cell architecture, as the changes observed in cell volume, valve thickness and sinking rates were minor compared to those caused by temperature (Table 1). The

increased incorporation rates of BSi also helped to preserve the properties of the diatom cell wall. Because we observed the effects of increased temperature and $p\text{CO}_2$ on the rate of silica incorporation by diatoms to be opposite, we were curious to elucidate how both stressors acting together will influence silica incorporation rates in the future. We applied the Independent Action model, as described in Payne et al., (2001); this model tests for joint additive effects of independently acting

stressors (i.e. pollutants). If the two stressors combined for an additive effect, this model predicted the addition to be negative (Fig. 3A). However, the joint effect observed here deviated from additivity,





showing an interaction (Fig. 3C). When we increased the temperature to 6.7 ºC, we observed a significant ($p<0.05$) antagonistic effect of $p\mathrm{CO_2}$, as increased $p\mathrm{CO_2}$ caused a significant increase in silicification rates ($p<0.05$) (Fig. 3B). However, at 10.3 ºC, the highest temperature tested, the increase in $p\mathrm{CO_2}$ acted synergistically with temperature, resulting in a sharper decrease than was expected from

simple addition ($p<0.05$) (Fig. 3A and B). This indicates that the positive effect of increased $p\mathrm{CO_2}$ on incorporation rates might not be strong enough to counter the negative effect of the high temperatures tested here, particularly at intermediate $p\mathrm{CO_2}$ values. This temperature dependence of the effects of increased $p\mathrm{CO_2}$ has also been observed for the gross primary production of a community of phytoplankton from the European Arctic (Holding et al., 2015).

Cell size and density are, together with the physiological state of the diatom, key factors influencing cell buoyancy and determining sinking rates (U) (Miklasz and Denny, 2010). Stokes' law predicts that if cell density remains constant, the sinking speed of a cell will be directly correlated to its radius. Thus, the decrease in cell size observed under warming (Coello-Camba et al., 2014 and 2015) is expected to

reduce the sinking rates of diatoms. We observed this trend in our results, but only with statistical significance for the largest diatom (21.4 µm diameter, $p < 0.05$) (Table 3).
The scaling relationship between sinking speed and diatom size can be affected by variations in valve thickness (Miklasz and Denny, 2010) as silica is denser than organic carbon (Smetacek, 1985). According to this, the sinking rates obtained here showed very slight decreasing trends with increased

temperature, in contrast with the potential decrease in U expected due to non-conservative thickness. This buffering effect was most of all effective for the smallest diatoms observed here (6.6 and 7.4 µm in diameter), that showed significant differences ($p<0.05$) when the actual decrease in sinking rates was compared to the potential decrease expected without preserving valve thicknesses (data not shown).

Our results confirm that increased temperatures and $p\mathrm{CO_2}$ will have implications for key organisms, such as diatoms. We have found that the negative effects of warming may be counteracted by the effects of increased $p\mathrm{CO_2}$ for the benefit of diatom survival. This antagonistic effect may explain why at present the relative abundance of diatoms is increasing while that of other groups is decreasing (Hinder et al., 2012). The favorable effect of a lower pH environment, derived from higher $p\mathrm{CO_2}$, agrees with

the conditions estimated during the Cretaceous Period (Zeebe, 2001) when diatoms first appeared on Earth.
       A reduction in body size at both organism and community levels is a universal ecological response by ectothermic aquatic organisms facing global warming (Daufresne et al., 2009); this response has previously been identified for Arctic diatoms (Montagnes and Franklin, 2001; Atkinson et

al., 2003). At the community level in the Arctic Ocean, small-sized phytoplankton species are expected to thrive, while the number of larger species is predicted to decrease (Coello-Camba et al., 2004 and 2015). Increased $p\mathrm{CO_2}$ may contribute to the preservation of the valve thicknesses in these smaller cells, causing an increase in silicification rates that allows diatoms to maintain the protective properties of their valves despite the overall reduction in cell size. Besides buffering the damage incurred from higher

temperatures, the positive effect of higher levels of $p\mathrm{CO_2}$ on the silicification of the valves may also ensure that the sinking properties of diatoms are maintained. This is key for diatom survival, as they





keep a suitable retention time in the water column (Smetacek, 1985), but also for maintaining the current flux of silica and carbon to the deep ocean.

**5. Conclusions**

Diatoms are the major component of the spring bloom and the primary algal class exported from the photic zone in the Arctic (Hodal et al., 2012; Baumann et al., 2014), dominating productive areas in the oceans (Agustí et al., 2015). Our results demonstrate that the effects of increased temperature and $pCO_2$ on the silicification process in diatoms are interactive rather than additive, showing a temperature

dependent capacity of increased $pCO_2$ to buffer the negative effects of warming. Therefore, as long as the increase in temperature does not surpass the buffering capacity of $pCO_2$, the increase of this latter stressor will help diatoms to retain their sinking properties, preserving their role in the biogeochemical cycles of key elements, such as silica and carbon.

**Author contributions**

A.C.-C. and S.A. were responsible for running all experiments, performing data analysis and for writing and editing the manuscript.

**Competing Financial Interests statement**

The authors declare that they have no conflict of interest.

**Acknowledgments**

This research was supported by the project Arctic Tipping Points (ATP, contract # 226248) from the European Union. A. Coello-Camba was supported by a grant BES-2007-15193 from the Spanish
Ministry of Science and Innovation. We thank project coordinators C.M. Duarte and P. Wassmann, the crew of R/V Jan Mayen and Viking Explorer for assistance with sampling, and The University Centre in Svalbard (UNIS) for their hospitality.



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

**Figure captions**

**Fig. 1. Valve architecture and cell size in Arctic diatoms. A) Microphotograph of a clean diatom valve, where VT: valve thickness and CD: cell diameter. The relationship between B) cell volume (CV, μm$^3$) and C) valve thickness (VT, nm) with increased temperature for the most abundant groups of diatoms in Arctic communities.**

**Fig. 2. Effect of increased temperature and $p$CO$_2$ on BSi incorporation rates. These rates were measured during the last day of incubation of the ATP 2010 experiment. Plot A shows the effect of increased temperature and plot B shows the effect of increased $p$CO$_2$.**

**Fig. 3. Effects of temperature and $p$CO$_2$ alone or combined. A) Increased temperature alone (T(1)=6.7 ºC; T(2)=10.3 ºC ) decreased
silica incorporation rates (red columns); increased $p$CO$_2$ alone (CO$_2$(1)= 780.8 ppm; CO$_2$(2)=1652 ppm) increased silica**





incorporation rates (blue columns). B) The Independent Action model allowed us to estimate the expected combined effects of both stressors on silica incorporation rates and to predict an overriding effect of temperature, as defined by the decrease in incorporation rates represented here. C) Combined effects of increased temperature and $pCO_2$ observed on silica incorporation rates. At 6.7 ºC, increased $pCO_2$ can easily counter the negative effects of temperature, leading to increased incorporation rates. At

higher temperatures (10.3 ºC), this buffering effect of increased $pCO_2$ is not strong enough and a steep decrease in the incorporation rates takes place. All the results are expressed relative to the silica incorporation rates in the control (T=1.8 ºC, $pCO_2$= 217.7 ppm).









**Tables**

**Table 1. Experiment information.**

| Experiment | Sampling location | Experiment dates | Sampling water T (ºC) | Nominal temperature range tested (ºC) | Nominal $pCO_2$ tested (ppm) |
|---|---|---|---|---|---|
| **ATP 2009** | | | | | |
| **Open sea** | **SE of Svalbard** | **1-10 July** | **-1.19** | **1.5 to 10.5** | **-** |
| **Fjord** | **Isfjorden** | **10-19 July** | **6.2** | **1.5 to 10.5** | **-** |
| **ATP 2010** | **Isfjorden** | **24 June-8 July** | **1.4** | **1 to 10** | **380 and 1000** |





**Table 2. Mean incubation temperatures (ºC) measured along each experiment. Standard error (± 0.1).**

| ATP 2009: Open sea experiment (ºC) | ATP 2009: Fjord experiment (ºC) | ATP 2010 experiment (ºC) |
|---|---|---|
| 1.6 | 1.2 | 1.8 |
| 2.6 | 3.0 | 6.7 |
| 4.5 | 4.1 | 10.3 |
| 5.5 | 5.5 | |
| 7.6 | - | |
| 8.5 | 8.3 | |
| 10.5 | 10 | |



**Table 3. The mean ± SE effects of experimentally increased temperatures and $pCO_2$ on valve thickness, cell volume, and sinking speed. Asterisks mark statistically significant results (* $p < 0.05$; ** $p < 0.001$).**

| | | Temperature | $pCO_2$ |
|---|---|---|---|
| **Cell volume (CV)** | | $\mu m^3\ {}^\circ C^{-1}$ | $\mu m\ ppm^{-1}$ |
| | **Diatoms 21.4 μm** | -213.1 ± 108.2 * | |
| | **Diatoms 7.4 μm** | -3.82 ± 1.47 * | |
| | **Diatoms 6.6 μm** | -1.74 ± 0.81 * | -0.006 ± 0.005 ns |
| **Valve thickness (VT)** | | $nm\ {}^\circ C^{-1}$ | $nm\ ppm^{-1}$ |
| | **Diatoms 21.4 μm** | -0.72 ± 0.88 ns | |
| | **Diatoms 7.4 μm** | 0.76 ± 0.37 * | |
| | **Diatoms 6.6 μm** | 0.51 ± 0.26 ns | -0.04 ± 0.004 ** |
| **Sinking rates (U)** | | $md^{-1}\ {}^\circ C^{-1}$ | $md^{-1}\ ppm^{-1}$ |
| | **Diatoms 21.4 μm** | -0.04 ± 0.02 * | |
| | **Diatoms 7.4 μm** | -0.002 ± 0.001 ns | |
| | **Diatoms 6.6 μm** | -0.001 ± 0.001 ns | $-5.4 \times 10^{-5} \pm 6.3 \times 10^{-6}$ ** |



Figure 1

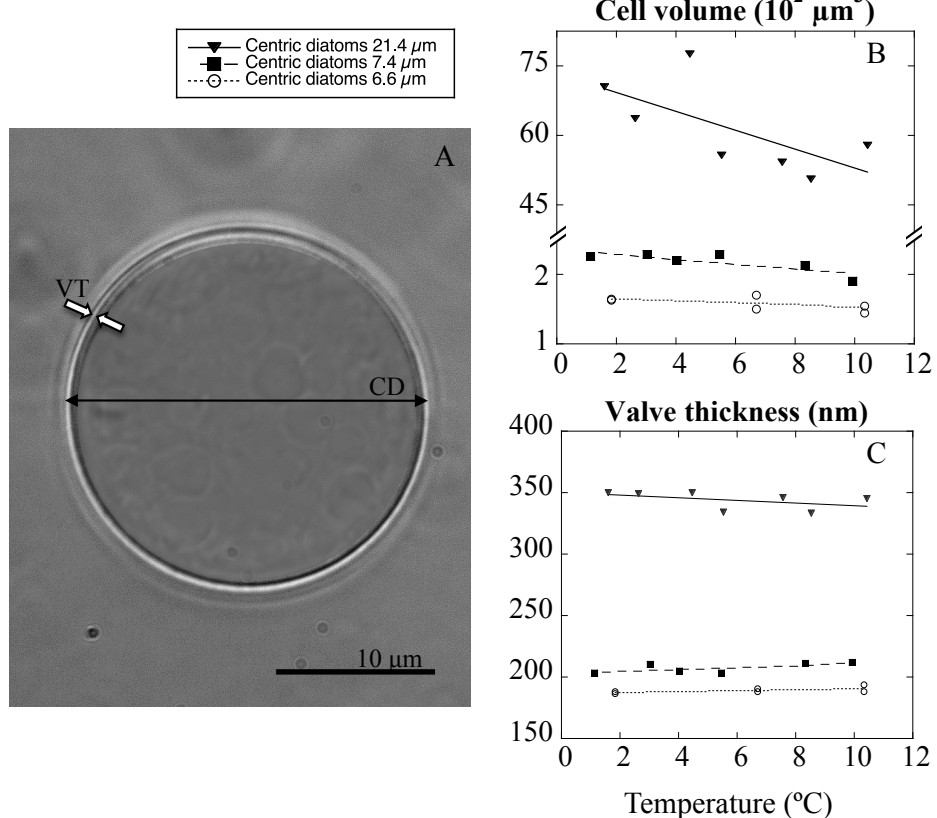




Figure 2

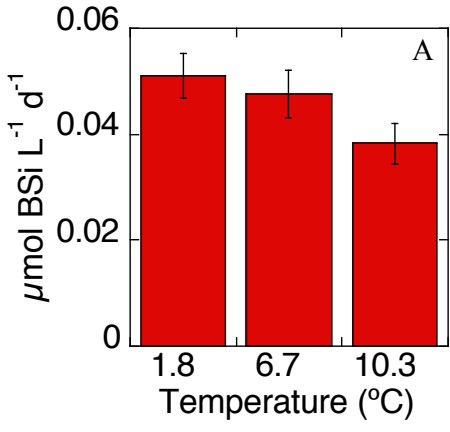
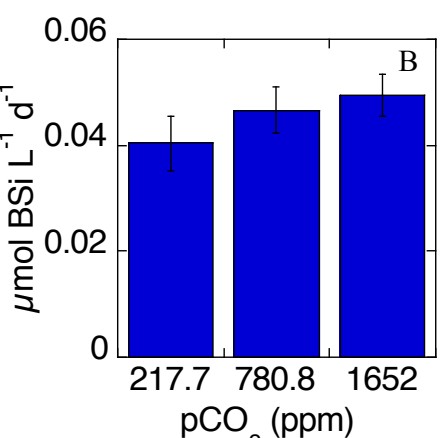




Figure 3

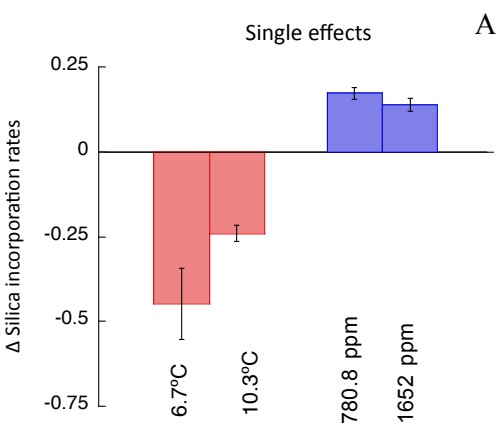

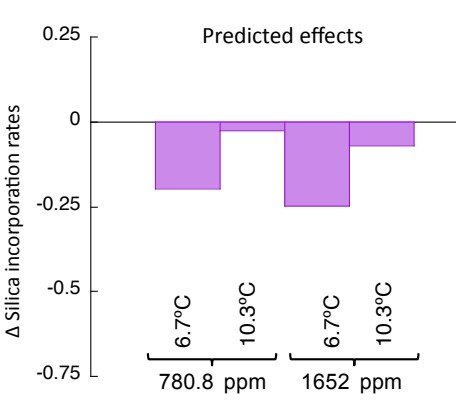

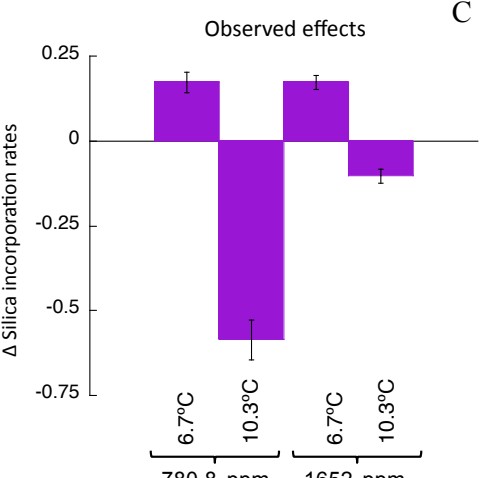