# Peer review of "Acidification counteracts negative effects of warming on diatom silicification"

_Biogeosciences, 2016_

## Referee Comment (RC1) · Anonymous Referee #1 · 22 Nov 2016

Coello-Camba and Agusti describe the results of three experiments, testing the effect of temperature and pCO2 on diatom cell volume and valve thickness. I do not believe that they have tested the actual effect on diatom silicification as stated in the title for reasons that I explain further down in this review. The topic of how climate change related environmental drivers affect phytoplankton physiology and thereby possibly their ability to protect themselves against grazers is a very important one. However, I feel that this manuscript lacks important information, especially in the methods section and I am not convinced that the experimental set up and some of the methods used and especially the amount of data analysed are appropriate to allow the authors to draw the conclusions they did. I therefore do not recommend publication of this manuscript

unless the authors can clarify all the issues addressed in detail below.

Methods: Page 2, Line 32: You state that in the 2010 experiment you had two pCO2 treatments at 380 and 1000 ppm. However, in the figure 2 there are three pCO2 treatments at 217.7 (which would be pre-industrial), 780.0 and 1652 ppm which would both be future scenarios. In figure 3 you only show 780.8 and 1652 ppm. Which one is true? If you did not have and ambient control treatment at 380 ppm please explain why.

Page 2, Line 35: How many 20l bottles were incubated? Only one per treatment without replication?

Page 2, Line 38-Page 3 Line 5: Were the bottles constantly bubbled during incubation or was the target pCO2 just adjusted in the beginning? If yes, it will have changed during incubation, was pH monitored?

Page 3, line 8: 200 $\mu$mol photons constant light seems very high. Can you show environmental data to show that these are average light intensities phytoplankton is exposed to in Arctic summer considering the deep mixing?

Page 3 line 10 following: My major concern about the cell size and valve thickness measurements is that I don't know how you can be sure that you measured individual frustules from the same species. Following the cleaning procedure, the cells will be empty and broken. Were you able to identify the species or at least genus? If yes please report them. If not how can you be sure that you did not measure cells from different species in each treatment? Also how many cells did you measure from each treatment? In figure 1 it seems like you only took one measurement per treatment as there are no error bars. If this is true, I do not believe that your data show any temperature trend but simply the natural variability in cell size that you find in any diatom species. In line 27 and 28 you state that you determined cell volume from the closest geometric figure. Did you measure the frustule height for each measured cell or did you just use the same estimated height for all measurements throughout

all treatments? If you do not have the exact cell height for each cell you measured your cell volume estimations will be very inaccurate. Cell height in diatoms is much more variable than frustule diameter even within the same species. At least you would need to know which species you have measured in each treatment and use an average literature value for its cell height.

Page 3 line 30 following: If I understand this correctly you used the PDMPO uptake over 24 hours I sub-incubations to determine the silicification in your treatments. But you do not describe any biomass measurements to relate to PDMPO uptake, do you just assume that growth rates and grazing were exactly the same in all treatments? Total PDMPO uptake does not say anything about silicification if you don't know the diatom biomass. The lower Si uptake at high temperatures could also mean that less diatoms grew in total, or more of less silicified diatom species but it does not necessarily mean that each diatom species changed its silicification.

Page 4, Line 23 following: I don't really understand why you are using this model if the assumption that temperature and pCO2 have additive effects is clearly wrong and does not fit your data. I don't see how this is helpful.

Page 5, Line 1 following: The estimation of sinking rates does only apply for dead cells as you assume constant cytoplasm density for all species and throughout all treatments. This should be made clear here and later in the results and discussion.

Page 5, Line 18-20: This should go in the results section

Page 5, Line 27,28: I hope the pH values did not vary between 8.2 and 7.5 which would be massive but that these are indeed the pH values for the three individual pCO2 treatments. Please state how and when pH was measured and if these are the start values or average over the course of the incubation.

Tables 1 and 2: I think these two tables should be combined. Also in table 1 you state that the temperatures for the first two experiments were 1.5 – 10.5 and from 1 – 10 in

experiment 3 but in table 2 temperature goes from 1.6 to 10.5 and from 1.2 to 10 in the first two experiments and from 1.8 to 10.3 in exp. 3. Also pCO2 in table 1 is once again described as 380 and 1000ppm. Please make sure you report the correct values throughout.

Table 3: I cannot believe that all the cells you measured had the exact same size (e.g. 21.4 $\mu$m in exp.1). There is always a natural variance in cell size in every species and in this study it is absolutely crucial to know what the natural variability is in order to be able to estimate changes in cell size caused by the different treatments. So please report the number of individual cells you measured in each treatment and the actual cell size of each.

Figure 1: From this figure I am not convinced that you were able to perform the measurements of valve thickness and cell diameter with the precision that you report. In table 3 you have calculated a change in valve thickness of less than 1 nM per °C. As the temperature range you have tested is 10°C this would mean that you have actually measured a 10 nM difference in valve thickness between these two treatments using light microscopy! I find this hard to believe.

---

## Referee Comment (RC2) · Anonymous Referee #2 · 9 Dec 2016

This paper deals with physiological response of Arctic diatoms to increasing of temperature and pCO2. Cell volume, valve thickness and silicon incorporation rate of diatoms were examined by using the natural diatom community. The authors showed that cell volume and valve thickness of diatoms were decreased as increase of temperature and pCO2, while silicon incorporation rates were increased. The authors described that increase CO2 and water temperature affect negative effect of diatom silicification. It has taken great efforts to incubate many large bottles in this study. Also, there is new information on silicon incorporation rates of diatoms using a novel fluorescence bye PDMPO. However, this manuscript contains significant problems in experimental designs and methods they used. Therefore, data the authors analyzed do not support

their conclusion and are inadequate to lead their conclusion. I suggest the current manuscript doesn't merit to be published. Please clarify my questions listed below.

Introduction P2 line 14-23. My main concern is that the reason why they used the natural phytoplankton community is unclear. In the introduction section the authors describes that they analyzed the effects of temperature and pCO2 on cell and valve dimensions and silicification, and possible interactions. For such purpose incubation experiments using unialgal strains in laboratory under severely controlled condition are appropriate to demonstrate physiological response of diatoms to environmental changes. Advantages using natural plankton community are to evaluate ecosystem responses, such as competition among the other phytoplankton, species succession among diatoms and effect of grazing by microzooplankton.

Methods

P2 line 33-P3 line 9. The authors should describe when, where and how they obtained water samples for general readers.

P2 line 31-32. While setup conditions of pCO2 were described to be 380 ppm and 1000 ppm in the materials and method section, the results at 217.7 ppm, 780.8 ppm and 1652 ppm were shown. Which is right?

P3 line 8. Was the setup condition of light intensity ($200\mu$mol, continuous light) appropriate? At least the authors should show daily PAR at the same latitude for reference.

P3 line 13. Why were the concentrated water samples frozen? This procedure would damage diatom frustules.

P3 line 14-15. Were the density of dead (empty) diatom cells checked before cleaning procedure?

P3 line 27-28. How did the authors measure height of diatom frustules to the pervalvar axis to calculate cell volume?

P3 line 38. The authors used 250 mL subsamples from 20L incubation bottles for measuring silicon incorporation rates. Did the authors conduct CO2 bubbling or monitor pH value in the subsamples?

Results

P6 line 10-15. I can't understand this sentence. Were the cell volumes of the centric diatoms 21.4$\mu$m decreased, although their diameters did not change? This means that height of diatoms to the pervalvar axis. Please clarify this more detail.

Discussion

P7 line 2-10. I disagree this conclusion. It is unclear whether size of the same species was decreased or the dominant diatoms successed from large species to small species because diatoms were not identified to species.

P7 line 24-26. I disagree this conclusion. It is possible that a decrease in silica incorporation rates was due to lower abundance of total diatoms at higher temperature. The authors should show the initial concentrations of biogenic silica or total biomass of diatoms in 250 mL subsamples before incubation with PDMPO.

P7 line 27-32. I disagree this conclusion. It is possible that an increase in silica incorporation rates was due to higher abundance of total diatoms at higher pCO2. The initial concentrations of biogenic silica or total biomass of diatoms in 250 mL subsamples should be shown.

Table 1 Please show the longitudes and latitudes at sampling locations.

Fig. 2A. Is this figure the result at 380 ppm or 1000 ppm? Please describe more details in the figure caption.

Fig. 2 B. Is this figure the result at 1.8°C, 6.7°C or 10.3°C? Please describe more details in the caption.

[Figure]

---

## Referee Comment (RC3) · Anonymous Referee #3 · 12 Dec 2016

General comments: This manuscript presents data on the effects of temperature and $CO_2$ on cell size, valve thickness, sinking rate and silica incorporation rate of in situ diatom communities. It's interesting to see that increased $CO_2$ mitigates the negative effects of warming on silicification. However, I found several serious problems in the study: 1) my main concern is the replicates in the experiment, no detailed information can be found in the manuscript. From figure 1, there is only one data point for one temperature treatment. 2) The carbonate system parameters are missing to further constrain carbonate chemistry. 3) I think it's not proper to classify species according to cell size. Cell size can vary a lot even for the same species. The dominate species information should be provided. 4) Why the silica incorporation rate normalized to

volume rather than biomass? If the biomass in different treatments were distinct, the rates can say nothing.

Specific comments: Page 1 line 1: I don't think this title is appropriate for this paper. In two of three experiments, the authors only focus on the effects of temperature. Moreover, the authors discuss a lot on effects of temperature, rather than interactions of OA and temperature.

Page 2 line 20: "stressors"? Increased CO2 mitigates the negative effects of increased temperature. So can you call CO2 "stressor"?

Page 2 line 26: The information of dominate species in these communities should be added.

Page 2 line 31: Two pCO2 treatments? In Figure 2, you showed three pCO2 levels. Moreover, the pCO2 values are self-contradictory in method and results parts.

Page 2 line 35: How many replicates in the experiment? In fig. 1, only one data point for per treatment. Does this mean that there is only one bottle for per treatment?

Page 3 line 6: Were the bottles aerated throughout the experiment or stopped when target pH was achieved?

Page 3 line 8: The light tubes on the top or side of bottles? Did the author measure light in bottles?

Page 3 line 9: The carbonate system parameters are missing to further constrain carbonate chemistry.

Page 3 line 12: The information of filtration pressure should be added.

Page 3 line 14-21: It's better to add some references for this method.

Page 3 line 27: How many samples measured for one treatment? Again, how many replicates for per treatment?

Page 3 line 38: When did the author measure the rate of incorporation of silica? At the end of experiments? Samples were incubated under light or darkness?

Page 4 line 1: The information of filtration pressure should be added.

Page 4 line 22: From my perspective, this model is useless for the discussion. You can analyze the interaction of these two factors from fig. 3A and C.

Page 5 line 9: Median values of density of the cytoplasm and cell wall density were used for calculation the sinking rate. However, I think these parameters may be species-specific and influenced by treatment, such as temperature.

Page 5 line 23: More detailed data analysis information should be provided.

Page 5 line 29: These values were mean of each pCO2 treatment? Please add the standard deviation. In the method, you said there were two pCO2 levels.

Page 6 line 2: Can you tell whether the test cells belonged to one species or one genus according to their valves?

Page 6 line 3-5: I think it's not proper to classify species according to cell size. Cell size can vary a lot even for the same species. The dominate species information should be provided.

Page 7 line18-21: These sentences are repetition of the method section.

Page 9 line 8: Cautions should be taken to draw this conclusion: you only test the interaction of pCO2 and temperature for the third experiment. What will happen for the second one? The in situ temperature for the second experiment is 6.2 °C. Will the increased pCO2 counteracts negative effects of warming when temperature increases by 4 °C or more for diatoms in these waters? Base on the third experiment (at 10.3 °C, increased pCO2 acted synergistically with temperature), the answer may be "no". The author should add some discussion about this.

Page 9 line 12: I suggest to change "stressor" to "factor".

Page 16, table 3: Can the microscopic method test the minimal variation of valve thickness ($\sim$ 7 nm for temperature increasing 10 °C)?

Page 17 figure 1: Why only one data point for one temperature treatment? How many replicates in the experiment?

Page 17 figure 2: For panel A, what's the pCO2 treatment for every temperature column? Mean value of three pCO2 treatments. Same for panel B, what's the temperature treatment for every pCO2 column? Why the rate normalized to volume rather than biomass? If the biomass in different treatments were distinct, the rates can say nothing.

---

## Author Comment (AC1) · 20 Feb 2017

Coello-Camba and Agusti describe the results of three experiments, testing the effect of temperature and pCO2 on diatom cell volume and valve thickness. I do not believe that they have tested the actual effect on diatom silicification as stated in the title for reasons that I explain further down in this review. The topic of how climate change related environmental drivers affect phytoplankton physiology and thereby possibly their ability to protect themselves against grazers is a very important one. However, I feel that this manuscript lacks important information, especially in the methods section and I am not convinced that the experimental set up and some of the methods used and especially the amount of data analysed are appropriate to allow the authors to draw

the conclusions they did. I therefore do not recommend publication of this manuscript unless the authors can clarify all the issues addressed in detail below.

- Methods: Page 2, Line 32: You state that in the 2010 experiment you had two pCO2 treatments at 380 and 1000 ppm. However, in the figure 2 there are three pCO2 treatments at 217.7 (which would be pre-industrial), 780.0 and 1652 ppm which would both be future scenarios. In figure 3 you only show 780.8 and 1652 ppm. Which one is true? If you did not have and ambient control treatment at 380 ppm please explain why.
- Authors'R response: In agreement with the reviewer's observation we realize that this information was not well described. We referred to the planned treatments, although we should instead refer to the actual treatments. The methods were well described in Coello-Camba et al. (2014). According to this, we improved the description in the Methods section (lines 33-34, page 2): "Seven temperature treatments were set for the 2009 experiments and three temperature treatments in 2010; in this last experiment temperatures were combined with three pCO2 treatments (Table 1)", and (lines 13-14, page 6): "The average CO2 values actually measured along the experiment resulted in 217.7 (37), 780.8 (46) and 1652(72) ppm respectively". In figure 3, only the two higher CO2 values appear because, as indicated in the methods section (lines 14-15, page 5), we used the lowest pCO2 value as reference point in order to apply the IA method.

- Page 2, Line 35: How many 20l bottles were incubated? Only one per treatment without replication? - Authors' response: We added information on the number of replicate bottles in lines 37-38, page 2 of the revised manuscript: "For the 2009 experiments we used two replicate bottles for each treatment, and three replicates for each treatment in the 2010 experiment."

- Page 2, Line 38-Page 3 Line 5: Were the bottles constantly bubbled during incubation or was the target pCO2 just adjusted in the beginning? If yes, it will have changed during incubation, was pH monitored? - Authors' response: The bottles were incubated under a constant bubbling of air-CO2; this information has been added to the manuscript in lines 3-5, page 3: "Throughout the 2010 experiment the target pCO2

level was achieved by fitting each experimental 20 L bottle with a bubbling system connected to CO2 bottles and air mixture bottles. The gas mixture was continuously provided by mass flow controllers (model GFC17, Aalborg Instruments and Controls, Inc.) (...)". Besides this, total hydrogen ion concentrations (i.e., pH) and total alkalinity (TA) were measured daily along the experiment (information added in lines 10-11, page 3).

- Page 3, line 8: 200 $\mu$mol photons constant light seems very high. Can you show environmental data to show that these are average light intensities phytoplankton is exposed to in Arctic summer considering the deep mixing? - Authors' response: We chose this value based on the PAR measurements performed at noon in 22 Arctic stations on a previous cruise (July 2007). Using a PUV 2500 Biospherical radiometer, the average PAR value at 5 m deep was 146 $\mu$mol photons m-2 s-1, reaching a maximum value of 470 $\mu$mol photons m-2 s-1 on July 14th, and a mimimum of 45 $\mu$mol photons m-2 s-1on July 22th. We added this information to the revised manuscript on lines 13-16, page 3.

- Page 3 line 10 following: My major concern about the cell size and valve thickness measurements is that I don't know how you can be sure that you measured individual frustules from the same species. Following the cleaning procedure, the cells will be empty and broken. Were you able to identify the species or at least genus? If yes please report them. If not how can you be sure that you did not measure cells from different species in each treatment? Also how many cells did you measure from each treatment? In figure 1 it seems like you only took one measurement per treatment as there are no error bars. If this is true, I do not believe that your data show any temperature trend but simply the natural variability in cell size that you find in any diatom species. In line 27 and 28 you state that you determined cell volume from the closest geometric figure. Did you measure the frustule height for each measured cell or did you just use the same estimated height for all measurements throughout all treatments? If you do not have the exact cell height for each cell you measured

your cell volume estimations will be very inaccurate. Cell height in diatoms is much more variable than frustule diameter even within the same species. At least you would need to know which species you have measured in each treatment and use an average literature value for its cell height. - Authors' response: The measurements performed during our experiments were done on the most abundant groups, and they were clearly differentiated from other groups. The process of cell cleaning has been described as a necessary methodology in order to facilitate diatom identification and analysis using light microscopy (i. e. Identifying marine phytoplankton (Tomas, 1997); Phytoplankton identification manual (Verlecar & Desai, 2004)), so in fact this step facilitates diatom identification. Also, according to the reviewer's comment we have added to the revised manuscript the identification to genus level of the diatom groups studied here (lines 22-26, page 6), as Coscinodiscus sp. (21.4 0.38 $\mu$m initial cell diameter) from the 2009 open sea community experiment, and Thalassiosira sp. population 1 (7.4 0.04 $\mu$m initial cell diameter) from the 2009 fjord community experiment), and population 2 (6.6 0.04 $\mu$m initial cell diameter) from the 2010 experiment". - As the reviewer observed here, the error bars in Figure 1 were missing, so we have completed this figure by adding the error bars to the plots. - As indicated by the reviewer, the referred information is missing. We have added the following information on the revised version of the manuscript: " Centric diatoms are more likely to appear in the microscope slide on a valve view, so the measurement of cell heights was more difficult to get. This way, we used an estimation of the average cell heights for each group based on the measurements performed in Olenina et al. (2006)" in lines 2-5, page 4. Valve height is more conservative between species than their diameter. Olenina et al. (2006) showed that the range of Coscinodiscus spp. and Thalassiosira spp. diameters was larger than the range of their valve heights, that were very conservative.

- Page 3 line 30 following: If I understand this correctly you used the PDMPO uptake over 24 hours I sub-incubations to determine the silicification in your treatments. But you do not describe any biomass measurements to relate to PDMPO uptake, do you just assume that growth rates and grazing were exactly the same in all treatments?

Total PDMPO uptake does not say anything about silicification if you don't know the diatom biomass. The lower Si uptake at high temperatures could also mean that less diatoms grew in total, or more of less silicified diatom species but it does not necessarily mean that each diatom species changed its silicification. - Authors' response: To determine the silica incorporation rate we followed the standard procedure described in Leblanc and Hutchins (2005) and Shimizu et al. (2011). This measurement is an incorporation rate, a time-related parameter. We estimated the silica incorporation rates per unit of diatom biomass in the revised manuscript (values shown in lines 5-6, page 8), and observed that this ratio did not show any significant relationship with increased temperature or pCO2. Silicification is performed by active cells, although the measurement of biomass is not related to the state of the cells and includes no actively growing cells and a component of detritical biomass. Probably, the presence of non active cell biomass influenced the incorporation rate vs. biomass ratio and prevented us for finding clear responses of the ratio with increased temperature or pCO2. The incorporation rates showed here reflecting the silicification process help us to identify the overall silicification responses of the communities and thus the consequences for the biogeochemical cycles.

- Page 4, Line 23 following: I don't really understand why you are using this model if the assumption that temperature and pCO2 have additive effects is clearly wrong and does not fit your data. I don't see how this is helpful. - Authors' response: The simple and intuitive interpretation of joint effects is only appropriate under the condition of a linear relationship between the intensity of the single stress factors and their effects (Coors and De Meester, 2008). Thus, the summation of effects could not be applied here as increased temperature and pCO2 typically show non-linear dose-response curves. The use of a mathematical model will help on the identification of the degree of interaction. The independent action (IA) model used here assumes additivity, and denotes synergy and antagonism by positive and negative significant deviations of the observed relative to the predicted effect, respectively, as described in Payne et al. (2001). This model is widely used in toxicology and has been successfully applied to test the interaction between stressors as toxic chemicals in zooplankton (Coors & De Meester, 2008; Carreja et al., 2016). Moreover, this method allows a good graphic representation of the results. We have added in the revised manuscript a better explanation of the usefulness of the model applied (lines 1-6, page 5). "In order to determine mathematically the existence of synergy or antagonism in the effects of increased temperature and pCO2 in silica incorporation rates of diatoms, we used the independent action (IA) model described by Payne et al. (2001). This model has been recommended for the prediction of the joint effects of dissimilarly acting factors (stressors that influence independently the regulation of a life-history trait by different mechanisms). It assumes additivity and denotes non-additivity by deviations of the measured from the predicted (reference) responses (Coors and De Meester, 2008)".

- Page 5, Line 1 following: The estimation of sinking rates does only apply for dead cells as you assume constant cytoplasm density for all species and throughout all treatments. This should be made clear here and later in the results and discussion.
- Authors' response: We improved the description in the revised manuscript. The lack of cytoplasm density measurements in the literature compels us to use a constant value for living cells (Miklasz and Denny (2010). We used the same assumption in the formula to estimate the potential sinking rates of living diatoms. In the revised version of the manuscript, in lines 36-37, page 5, we added the following information: " We have assumed here a constant value for cytoplasm and valve densities in living diatoms as there are very few literature data on this topic (Miklasz and Denny, 2010)" and it is not clear yet if the cytoplasm density could be species-specific (Miklasz and Denny, 2010). We used the equation described in Miklasz and Denny (2010) to estimate the changes in the potential sinking speed of a diatom due to variations in valve size and thickness.

- Page 5, Line 18-20: This should go in the results section - Authors' response: As suggested by the reviewer, the referred sentence has been relocated to the Results section (lines 17-18, page 6).

- Page 5, Line 27,28: I hope the pH values did not vary between 8.2 and 7.5 which

would be massive but that these are indeed the pH values for the three individual pCO2 treatments. Please state how and when pH was measured and if these are the start values or average over the course of the incubation. - Authors' response: These pCO2 values for each treatment have been obtained from daily measurements, as indicated on a previous response: " Total hydrogen ion concentrations (i.e., pH) and total alkalinity (TA) were measured daily along the experiment (see Coello-Camba et al. 2014)"(lines 10-11, page 3). This information has been clarified in lines 14-16, page 6: "The pH values obtained for each pCO2 treatment by averaging daily measurements of the last week of incubation were 8.2 (0.1), 7.9 (0.1) and 7.5 (0.0) pH units ".

- Tables 1 and 2: I think these two tables should be combined. Also in table 1 you state that the temperatures for the first two experiments were 1.5 – 10.5 and from 1 – 10 in experiment 3 but in table 2 temperature goes from 1.6 to 10.5 and from 1.2 to 10 in the first two experiments and from 1.8 to 10.3 in exp. 3. Also pCO2 in table 1 is once again described as 380 and 1000ppm. Please make sure you report the correct values throughout. - Authors' response: According to the reviewer's suggestion, both tables have been combined (new Table 1).

- Table 3: I cannot believe that all the cells you measured had the exact same size (e.g. 21.4 _m in exp.1). There is always a natural variance in cell size in every species and in this study it is absolutely crucial to know what the natural variability is in order to be able to estimate changes in cell size caused by the different treatments. So please report the number of individual cells you measured in each treatment and the actual cell size of each. - Authors' response: We added more information to indicate the natural variability in the cell size of the diatoms described here in the revised version of the manuscript. To do so, we added the standard errors in the measurements of cell valve diameters (lines 23-26, page 6), as Coscinodiscus sp. (21.4 0.38 $\mu$m initial cell diameter) from the 2009 open sea community experiment, and Thalassiosira sp. population 1 (7.4 0.04 $\mu$m initial cell diameter) from the 2009 fjord community experiment), and population 2 (6.6 0.04 $\mu$m initial cell diameter) from the 2010 experiment. We also

added in the methods section of the revised manuscript the minimum number of cells measured per treatment (lines 26-27, page 6).

- Figure 1: From this figure I am not convinced that you were able to perform the measurements of valve thickness and cell diameter with the precision that you report. In table 3 you have calculated a change in valve thickness of less than 1 nM per _C. As the temperature range you have tested is 10_C this would mean that you have actually measured a 10 nM difference in valve thickness between these two treatments using light microscopy! I find this hard to believe. - Authors' response: Although the theoretical maximum resolving power in optical microscopy is 0.2 $\mu$m at 1000x magnification, this parameter is quite dependent on the detection mode used. Digital imaging systems allow image enhancement and perform considerably better contrast than the nonlinear human eye, so the standard resolution criteria do not apply when these image analysis softwares are used (Hajjar et al., 1999). This way, the method we followed here used 1600x magnification allowing a maximum resolving power of 0.125 $\mu$m, plus image analysis system, had an adequate resolution (0.05 $\mu$m approx.) to perform the measurements of valve thicknesses. Besides this, as we indicated in the text, we did not observe significant variations in the valve thickness measurements with temperature or pCO2 (lines 34-35, page 6). We added the methodological information in the revised manuscript lines 36-39, page 3.

Figure 1

[Figure]

**Fig. 1.** New Figure 1

Table 1

| | Experiment | | |
|---|---|---|---|
| | ATP 2009 | | ATP 2010 |
| | Open sea | Fjord | |
| Sampling location | SE of Svalbard | Isfjorden | Isfjorden |
| Latitude/Longitude | 77°N / 28°E | 78°N / 14°E | 78°N / 13°E |
| Experiment dates | 1-10 July | 10-19 July | 24 June-8 July |
| Sampling water T (°C) | -1.19 | 6.2 | 1.4 |
| | 1.6 | 1.2 | 1.8 |
| | 2.6 | 3 | |
| Mean incubation | 4.5 | 4.1 | |
| T measured | 5.5 | 5.5 | |
| (°C, ±0.1) | 7.6 | - | 6.7 |
| | 8.5 | 8.3 | |
| | 10.5 | 10 | 10.3 |
| Mean $pCO_2$ | | | 217.7 (±37) |
| values measured | - | - | 780.8 (±46) |
| (ppm, ±SE ) | | | 1652 (±72) |

**Fig. 2.** New Table 1

---

## Author Comment (AC2) · 20 Feb 2017

This paper deals with physiological response of Arctic diatoms to increasing of temperature and pCO2. Cell volume, valve thickness and silicon incorporation rate of diatoms were examined by using the natural diatom community. The authors showed that cell volume and valve thickness of diatoms were decreased as increase of temperature and pCO2, while silicon incorporation rates were increased. The authors described that increase CO2 and water temperature affect negative effect of diatom silicification. It has taken great efforts to incubate many large bottles in this study. Also, there is new information on silicon incorporation rates of diatoms using a novel fluorescence bye PDMPO. However, this manuscript contains significant problems in experimental

designs and methods they used. Therefore, data the authors analyzed do not support their conclusion and are inadequate to lead their conclusion. I suggest the current manuscript doesn't merit to be published. Please clarify my questions listed below.

- Introduction P2 line 14-23. My main concern is that the reason why they used the natural phytoplankton community is unclear. In the introduction section the authors describes that they analyzed the effects of temperature and pCO2 on cell and valve dimensions and silicification, and possible interactions. For such purpose incubation experiments using unialgal strains in laboratory under severely controlled condition are appropriate to demonstrate physiological response of diatoms to environmental changes. Advantages using natural plankton community are to evaluate ecosystem responses, such as competition among the other phytoplankton, species succession among diatoms and effect of grazing by microzooplankton. - Authors' response: The main reason why we use natural phytoplankton communities here is to test the actual responses of Arctic communities. Laboratory data cannot reflect the complexity of the biotic and abiotic interactions that take place in nature and in particular in Arctic waters. Silicification processes have been demonstrated to be influenced by the concentration of silica present in the medium, influencing the amount of silica eventually incorporated to the cell (Finkel et al. 2010). Moreover, different environmental factors acting simultaneously are difficult to reproduce in the laboratory. Here our interest was focused on the process of silicification in arctic communities, as affected by increased levels of temperature and pCO2. This study focuses more on the biogeochemical consequences than on the diatom physiology alone. The use of cultures will imply a strong simplification of the environment.

- Methods: P2 line 33-P3 line 9. The authors should describe when, where and how they obtained water samples for general readers. - Authors' response: We improved the description of the sampling with this information included now in the new Table 1. In lines 25-29, page 2 of the new manuscript we refer the reader to the previous publications where these experiments were described in more detail, " Three incubation

experiments were conducted in the Arctic Ocean using natural phytoplankton communities sampled at the surface waters. These experiments were run at the University Centre in Svalbard (UNIS) in Longyearbyen during the summers of 2009 and 2010 (see Coello-Camba et al., 2014 and 2015). Information on sampling location, dates and conditions of temperature and pCO2 treatments are also described in new Table 1".

- P2 line 31-32. While setup conditions of pCO2 were described to be 380 ppm and 1000 ppm in the materials and method section, the results at 217.7 ppm, 780.8 ppm and 1652 ppm were shown. Which is right? - Authors' response: In agreement with the reviewer's observation we realize that this information was not well described. We referred to the planned treatments, although we should instead refer to the actual treatments. The methods were well described in Coello-Camba et al. (2014). According to this, we improved the description in the Methods section (lines 33-34, page 2): "Seven temperature treatments were set for the 2009 experiments and three temperature treatments in 2010; in this last experiment temperatures were combined with three pCO2 treatments (Table 1)", and (lines 13-14, page 6): "The average CO2 values actually measured along the experiment resulted in 217.7 (37), 780.8 (46) and 1652(72) ppm respectively".

- P3 line 8. Was the setup condition of light intensity (200mol, continuous light) appropriate? At least the authors should show daily PAR at the same latitude for reference. - Authors' response: We chose this value based on the PAR measurements performed at noon in 22 Arctic stations on a previous cruise (July 2007). Using a PUV 2500 Biospherical radiometer, the average PAR value at 5 m deep was 146 $\mu$mol photons m-2 s-1, reaching a maximum value of 470 $\mu$mol photons m-2 s-1 on July 14th, and a mimimum of 45 $\mu$mol photons m-2 s-1on July 22th. We added this information to the revised manuscript on lines 13-16, page 3.

- P3 line 13. Why were the concentrated water samples frozen? This procedure would damage diatom frustules. - Authors' response: The samples were frozen to preserve

them until microscopic analysis on land. This procedure would not affect cell size or thickness, the main features to be measured here.

- P3 line 14-15. Were the density of dead (empty) diatom cells checked before cleaning procedure? - Authors' response: Yes, the abundance of empty diatoms was low. We checked the viability of some of the groups found, showing high viability along the experiment.

- P3 line 27-28. How did the authors measure height of diatom frustules to the pervalvar axis to calculate cell volume? - Authors' response: As indicated by the reviewer, the referred information is missing. We have added the following information on the revised version of the manuscript: " Centric diatoms are more likely to appear in the microscope slide on a valve view, so the measurement of cell heights was more difficult to get. This way, we used an estimation of the average cell heights for each group based on the measurements performed in Olenina et al. (2006)" in lines 2-5, page 4. Valve height is more conservative between species than their diameter. Olenina et al. (2006) showed that the range of Coscinodiscus spp. and Thalassiosira spp. diameters was larger than the range of their valve heights, that were very conservative.

- P3 line 38. The authors used 250 mL subsamples from 20L incubation bottles for measuring silicon incorporation rates. Did the authors conduct $CO_2$ bubbling or monitor pH value in the subsamples? - Authors' response: No, the $CO_2$ levels were monitored along all the experiments, but for the PDMPO subsamples this was not feasible. We, as other authors (i.e. Sugie et al., 2013), followed a standard procedure to perform these measurements in $pCO_2$ experiments. It is not necessary to bubble $CO_2$ into the PDMPO bottles as the relatively low biomass present in these subsamples together with the short incubation time allows the assumption that the change in the carbonate chemistry of the bottles during this 24h incubation period was small (Sugie et al., 2013).

- Results: P6 line 10-15. I can't understand this sentence. Were the cell volumes of the centric diatoms 21.4m decreased, although their diameters did not change? This

means that height of diatoms to the pervalvar axis. Please clarify this more detail. - Authors' response: The referred paragraph has been modified (lines 31-32, page 6) to make more clear that the diameter decreased with temperature. We removed the quotation of cell diameter i.e. 21.4 $\mu$m to avoid misunderstandings. Now we refer to the different cell groups as Thalassiosira sp. populations 1 and 2 or Coscinodicus sp.

- Discussion: P7 line 2-10. I disagree this conclusion. It is unclear whether size of the same species was decreased or the dominant diatoms successed from large species to small species because diatoms were not identified to species. - Authors' response: We have added to the revised manuscript the identification to genus level of the diatom groups studied here (lines 23-26, page 6), as Coscinodiscus sp. (21.4 0.38 $\mu$m initial cell diameter) from the 2009 open sea community experiment, and Thalassiosira sp. population 1 (7.4 0.04 $\mu$m initial cell diameter) from the 2009 fjord community experiment), and population 2 (6.6 0.04 $\mu$m initial cell diameter) from the 2010 experiment. These groups were clearly differentiated from other less abundant groups and we can say that it is the size of these groups that decreased with temperature, instead of species succession from bigger to smaller species.

- P7 line 24-26. I disagree this conclusion. It is possible that a decrease in silica incorporation rates was due to lower abundance of total diatoms at higher temperature. The authors should show the initial concentrations of biogenic silica or total biomass of diatoms in 250 mL subsamples before incubation with PDMPO. - Authors' response: To determine the silica incorporation rate we followed the standard procedure described in Leblanc and Hutchins (2005) and Shimizu et al. (2011). This measurement is an incorporation rate, a time-related parameter. We estimated the silica incorporation rates per unit of diatom biomass in the revised manuscript (values shown in lines 5-6, page 8), and observed that this ratio did not show any significant relationship with increased temperature or pCO2. Silicification is performed by active cells, although the measurement of biomass is not related to the state of the cells and includes no actively growing cells and a component of detrital biomass. Probably, the presence of non

active cell biomass influenced the incorporation rate vs. biomass ratio and prevented us for finding clear responses of the ratio with increased temperature or pCO2. The incorporation rates showed here reflecting the silicification process help us to identify the overall silicification responses of the communities and thus the consequences for the biogeochemical cycles.

- P7 line 27-32. I disagree this conclusion. It is possible that an increase in silica incorporation rates was due to higher abundance of total diatoms at higher pCO2. The initial concentrations of biogenic silica or total biomass of diatoms in 250 mL subsamples should be shown. - Authors' response: See above.

- Table 1 Please show the longitudes and latitudes at sampling locations. - Authors' response: The suggested information has been added to Table 1.

- Fig. 2A. Is this figure the result at 380 ppm or 1000 ppm? Please describe more details in the figure caption. - Authors' response: In this figure we show the effect of temperature considering all pCO2 treatments.

- Fig. 2 B. Is this figure the result at 1.8C, 6.7C or 10.3C? Please describe more details in the caption. - Authors' response: In this figure we show the effect of temperature considering all temperature treatments.
* * *
Table 1

| | Experiment | | |
|---|---|---|---|
| | ATP 2009 | | ATP 2010 |
| | Open sea | Fjord | |
| Sampling location | SE of Svalbard | Isfjorden | Isfjorden |
| Latitude/Longitude | 77°N / 28°E | 78°N / 14°E | 78°N / 13°E |
| Experiment dates | 1-10 July | 10-19 July | 24 June-8 July |
| Sampling water T (°C) | -1.19 | 6.2 | 1.4 |
| | | | |
| | 1.6 | 1.2 | 1.8 |
| | 2.6 | 3 | |
| Mean incubation | 4.5 | 4.1 | |
| T measured | 5.5 | 5.5 | |
| (°C, ±0.1) | 7.6 | - | 6.7 |
| | 8.5 | 8.3 | |
| | 10.5 | 10 | 10.3 |
| | | | |
| Mean $p$CO$_2$ | | | 217.7 (±37) |
| values measured | - | - | 780.8 (±46) |
| (ppm, ±SE ) | | | 1652 (±72) |

**Fig. 1.** New Table 1

---

## Author Comment (AC3) · 20 Feb 2017

Alexandra Coello-Camba and Susana Agustí

Alexandra.camba@kaust.edu.sa

General comments: This manuscript presents data on the effects of temperature and $CO_2$ on cell size, valve thickness, sinking rate and silica incorporation rate of in situ diatom communities. It's interesting to see that increased $CO_2$ mitigates the negative effects of warming on silicification. However, I found several serious problems in the study: 1) my main concern is the replicates in the experiment, no detailed information can be found in the manuscript. From figure 1, there is only one data point for one temperature treatment. 2) The carbonate system parameters are missing to further constrain carbonate chemistry. 3) I think it's not proper to classify species according to cell size. Cell size can vary a lot even for the same species. The dominate species information should be provided. 4) Why the silica incorporation rate normalized to volume rather than biomass? If the biomass in different treatments were distinct, the rates can say nothing. - Authors' response: 1) More information on the number of replicates has been added (lines 37-38, page 2): " For the 2009 experiments we used two replicate bottles for each treatment, and three replicates for treatment in the 2010 experiment.". 2) More information on the carbonate parameters can be found on Coello-Camba et al. (2014) in the revised manuscript we added the following paragraph (lines 10-11, page 3: "Total hydrogen ion concentrations (i.e., pH) and total alkalinity (TA) were measured daily along the experiment (see Coello-Camba et al. 2014)"). 3) According to the reviewer's comment, we identified the diatom groups observed here to genus level (lines 22-26, page 6): " Valve measurements were performed on the centric diatoms most abundantly observed in our samples after a process of cell cleaning (Fig. 1A); these were identified to genus level as Coscinodiscus sp. (21.4 0.38 $\mu$m initial cell diameter) from the 2009 open sea community experiment, and Thalassiosira sp. population 1 (7.4 0.04 $\mu$m initial cell diameter) from the 2009 fjord community experiment), and population 2 (6.6 0.04 $\mu$m initial cell diameter) from the 2010 experiment". 4) The method followed here (Leblanc and Hutchins (2005) and Shimizu et al. (2011)) allows the calculation of the incorporation rates of biogenic silica in $\mu$mol BSi L-1 d-1units, as it is referred to the concentration of PDMPO incorporated in a specific volume of sample (250 mL) during one day of incubation. To determine the silica incorporation rate we followed the standard procedure described in Leblanc and Hutchins (2005) and Shimizu et al. (2011). This measurement is an incorporation rate, a time-related parameter. We estimated the silica incorporation rates per unit of diatom biomass in the revised manuscript (values shown in lines 5-6 page 8), and observed that this ratio did not show any significant relationship with increased temperature or pCO2. Silicification is performed by active cells, although the measurement of biomass is not related to the state of the cells and includes no actively growing cells and a component of detritical biomass. Probably, the presence of non active cell biomass influenced the incorporation rate vs. biomass ratio and prevented us for finding clear responses of the ratio

with increased temperature or pCO2. The incorporation rates showed here reflecting the silicification process help us to identify the overall silicification responses of the communities and thus the consequences for the biogeochemical cycles.

Specific comments: - Page 1 line 1: I don't think this title is appropriate for this paper. In two of three experiments, the authors only focus on the effects of temperature. Moreover, the authors discuss a lot on effects of temperature, rather than interactions of OA and temperature. - Authors' response: We have modified the title to include the negative effect of increased temperature, highlighting the main finding during our study: "Acidification counteracts negative effects of increased temperature on diatom silicification"

-Page 2 line 20: "stressors"? Increased CO2 mitigates the negative effects of increased temperature. So can you call CO2 "stressor"? - Authors' response: As the term stressor results confusing in this sentence, we have changed it for the word "factors" in the reviewed version of the manuscript (lines 21-23, page 2).

-Page 2 line 26: The information of dominate species in these communities should be added. - Authors' response: As suggested by the reviewer, the main groups studied here have been identified to a genus level (lines 22-26, page 6: " Valve measurements were performed on the centric diatoms most abundantly observed in our samples after a process of cell cleaning (Fig. 1A); these were identified to genus level as Coscinodiscus sp. (21.4 0.38 $\mu$m initial cell diameter) from the 2009 open sea community experiment, and Thalassiosira sp. population 1 (7.4 0.04 $\mu$m initial cell diameter) from the 2009 fjord community experiment), and population 2 (6.6 0.04 $\mu$m initial cell diameter) from the 2010 experiment".

-Page 2 line 31: Two pCO2 treatments? In Figure 2, you showed three pCO2 levels. Moreover, the pCO2 values are self-contradictory in method and results parts. - Authors' response: In agreement with the reviewer's observation we realize that this information was not well described. We referred to the planned treatments, although

we should instead refer to the actual treatments. The methods were well described in Coello-Camba et al. (2014). According to this, we improved the description in the Methods section (lines 33-34, page 2): "Seven temperature treatments were set for the 2009 experiments and three temperature treatments in 2010; in this last experiment temperatures were combined with three pCO2 treatments (Table 1)", and (lines 13-14, page 6): "The CO2 values actually measured along the experiment resulted in 217.7 (37), 780.8 (46) and 1652(72) ppm respectively".

-Page 2 line 35: How many replicates in the experiment? In fig. 1, only one data point for per treatment. Does this mean that there is only one bottle for per treatment? - Authors' response: More information on the number of replicates has been added (lines 37-38, page 2): " For the 2009 experiments we used two replicate bottles for each treatment, and three replicates for treatment in the 2010 experiment.". Also, Figure 1 has been completed by adding the error bars to the plots.

-Page 3 line 6: Were the bottles aerated throughout the experiment or stopped when target pH was achieved? - Authors' response: The bottles were constantly aerated throughout all the incubation time. This information has been added to lines 3-7, page 3:" Throughout the 2010 experiment the target pCO2 level was achieved by fitting each experimental 20 L bottle with a bubbling system connected to CO2 bottles and air mixture bottles. The gas mixture was continuously provided by mass flow controllers (model GFC17, Aalborg Instruments and Controls, Inc.), setting a flow rate of 0-10 L min-1 for air mixture and 0-10 mL min-1 for CO2".

-Page 3 line 8: The light tubes on the top or side of bottles? Did the author measure light in bottles? - Authors' response: The light tubes were located at the top of the incubation chambers. We had previously measured the light transmitted through poly-carbonate bottles, and observed that their walls filter UVB radiation and reduce a 10% of PAR.

Page 3 line 9: The carbonate system parameters are missing to further constrain carbonate chemistry. - Authors' response: We added more information on the carbonate parameters in the revised manuscript (lines 10-11, page 3: "Total hydrogen ion concentrations (i.e., pH) and total alkalinity (TA) were measured daily along the experiment (see Coello-Camba et al. 2014)").

Page 3 line 12: The information of filtration pressure should be added. - Authors' response: For all our filtrations we used low vacuum pressures. We specified in the revised manuscript that we used gentle filtration to process our samples avoiding cell damage (line 20, page 3 and line 18, page 4).

Page 3 line 14-21: It's better to add some references for this method. - Authors' response: As suggested by the reviewer, this information has been added to the revised manuscript (lines 22-25, page 3).

Page 3 line 27: How many samples measured for one treatment? Again, how many replicates for per treatment? - Authors' response: Each treatment has been sampled for cell measurements at the end of incubation, two replicates per treatment (lines 13-14, page 4: " Initial, intermediate and final samples (2 replicates of each) were taken to determine the rate at which newly synthesized silica was being incorporated into the valves of diatoms".

Page 3 line 38: When did the author measure the rate of incorporation of silica? At the end of experiments? Samples were incubated under light or darkness? - Authors' response: As indicated above, in lines 13-14, page 4 we indicate the timing of the samplings for the measurement of silica incorporation rates. The bottles for measuring this parameter were incubated under the same conditions than the correspondent treatment (lines 16-17, page 4): " 250 mL of sample were incubated with PDMPO (to a final concentration of 0.125 $\mu$M) for 24 h under the corresponding light, temperature and pCO2 conditions for each treatment".

Page 4 line 1: The information of filtration pressure should be added. - Authors' response: See above.

Page 4 line 22: From my perspective, this model is useless for the discussion. You can analyze the interaction of these two factors from fig. 3A and C. - Authors' response: By using this model we can define the presence or absence of interaction by comparison between observed (3C) and predicted effects (3B). We have added in the revised manuscript a better explanation of the usefulness of the model applied (lines 1-6, page 5). "In order to determine mathematically the existence of synergy or antagonism in the effects of increased temperature and $pCO_2$ in silica incorporation rates of diatoms, we used the independent action (IA) model described by Payne et al. (2001). This model has been recommended for the prediction of the joint effects of dissimilarly acting factors (stressors that influence independently the regulation of a life-history trait by different mechanisms). It assumes additivity and denotes non-additivity by deviations of the measured from the predicted (reference) responses (Coors and De Meester, 2008)". The use of simple and often intuitively applied effect summation is only appropriate under the condition of a linear relationship between the intensity of the single stress factors and their effects (Coors and De Meester, 2008). Thus, effect summation could not be applied here as increased temperature and $pCO_2$ typically show non-linear dose-response curves.

Page 5 line 9: Median values of density of the cytoplasm and cell wall density were used for calculation the sinking rate. However, I think these parameters may be species-specific and influenced by treatment, such as temperature. - Authors' response: We improved the description in the revised manuscript. The lack of cytoplasm density measurements in the literature compels us to use a constant value for living cells (Miklasz and Denny (2010). We used the same assumption in the formula to estimate the potential sinking rates of living diatoms. In the revised version of the manuscript, in lines 36-37, page 5, we added the following information: " We have assumed here a constant value for cytoplasm and valve densities in living diatoms as there are very few literature data on this topic (Miklasz and Denny, 2010)" and it is not clear yet if the cytoplasm density could be species-specific (Miklasz and Denny, 2010). We used the equation described in Miklasz and Denny (2010) to estimate the changes

in the potential sinking speed of a diatom due to variations in valve size and thickness.

Page 5 line 23: More detailed data analysis information should be provided. - Authors' response: This information has been added to line 8, page 6: "Student's t-tests were run to perform statistical analysis of data using JMP software"

Page 5 line 29: These values were mean of each pCO2 treatment? Please add the standard deviation. In the method, you said there were two pCO2 levels. - Authors' response: Yes; the methods section has been improved, and the suggested information has been added to the manuscript in lines 13-14, page 6: " The average CO2 values actually measured along the experiment resulted in 217.7 (37), 780.8 (46) and 1652 (72) ppm respectively".

Page 6 line 2: Can you tell whether the test cells belonged to one species or one genus according to their valves? - Authors' response: We could identify the diatom groups observed here to genus level. We added to the revised manuscript the identification to genus level of the diatom groups studied here (lines 23-26, page 6), as Coscinodiscus sp. (21.4 0.38 $\mu$m initial cell diameter) from the 2009 open sea community experiment, and Thalassiosira sp. population 1 (7.4 0.04 $\mu$m initial cell diameter) from the 2009 fjord community experiment), and population 2 (6.6 0.04 $\mu$m initial cell diameter) from the 2010 experiment.

Page 6 line 3-5: I think it's not proper to classify species according to cell size. Cell size can vary a lot even for the same species. The dominate species information should be provided. - Authors' response: See above.

Page 7 line18-21: These sentences are repetition of the method section. - Authors' response: As the reviewer stated here, the referred sentence was repeated from the Methods section and has been removed in the revised version of the manuscript.

Page 9 line 8: Cautions should be taken to draw this conclusion: you only test the interaction of pCO2 and temperature for the third experiment. What will happen for

the second one? The in situ temperature for the second experiment is 6.2 C. Will the increased pCO2 counteracts negative effects of warming when temperature increases by 4C or more for diatoms in these waters? Base on the third experiment (at 10.3, increased pCO2 acted synergistically with temperature), the answer may be " no". The author should add some discussion about this. - Authors' response: We have modified the referred paragraph in the new version of the manuscript (lines 2-10, page 10) "Our results demonstrate that the effects of increased temperature and pCO2 on the silicification process in the diatoms studied here are interactive rather than additive, showing a temperature dependent capacity of increased pCO2 to buffer the negative effects of warming. Therefore, as long as the increase in temperature does not surpass the buffering capacity of pCO2 (expected threshold above 6C (Holding et al., 2015) the increase of this latter factor will help diatoms to retain their sinking properties, preserving their role in the biogeochemical cycles of key elements, such as silica and carbon". Our results demonstrate that the effects of increased temperature and pCO2 on the silicification process in diatoms are interactive rather than additive, showing a temperature dependent capacity of increased pCO2 to buffer the negative effects of warming. We observed that at about 6C the effect of increased pCO2 is interactive, with synergy counteracting the effect of temperature. But further increases in temperature would be too strong to be balanced by pCO2, and their interaction would then be synergistic, leading to stronger decreases in the silica incorporation rates than those predicted by simple additivity.

Page 9 line 12: I suggest to change "stressor" to "factor". - Authors' response: As suggested by the reviewer, in this sentence in the reviewed version of the manuscript we changed the word "stressor" to "factor" (line 8, page 10).

Page 16, table 3: Can the microscopic method test the minimal variation of valve thickness ( 7 nm for temperature increasing 10C )? - Authors' response: Although the theoretical maximum resolving power in optical microscopy is 0.2 $\mu$m at 1000x magnification, this parameter is quite dependent on the detection mode used. Digital imaging
systems allow image enhancement and perform considerably better contrast than the nonlinear human eye, so the standard resolution criteria do not apply when these image analysis softwares are used (Hajjar et al., 1999). This way, the method we followed here used 1600x magnification allowing a maximum resolving power of 0.125 $\mu$m, plus image analysis system, had an adequate resolution (0.05 $\mu$m approx.) to perform the measurements of valve thicknesses. Besides this, as we indicated in the text, we did not observe significant variations in the valve thickness measurements with temperature or pCO2 (lines 34-35, page 6). We added the methodological information in the revised manuscript lines 36-39, page 3.

Page 17 figure 1: Why only one data point for one temperature treatment? How many replicates in the experiment? - Authors' response: The referred figure has been modified by adding the error bars to the points; as indicated above, (lines 37-38, page 2): " For the 2009 experiments we used two replicate bottles for each treatment, and three replicates for treatment in the 2010 experiment.".

Page 17 figure 2: For panel A, what's the pCO2 treatment for every temperature column? Mean value of three pCO2 treatments. Same for panel B, what's the temperature treatment for every pCO2 column? Why the rate normalized to volume rather than biomass? If the biomass in different treatments were distinct, the rates can say nothing. - Authors' response: In figure 2A all pCO2 treatments have been considered when analyzing the effect of temperature, and in figure 2B all temperature treatments have also been considered when analyzing the effect of pCO2. - The method followed here (Leblanc and Hutchins (2005) and Shimizu et al. (2011)) allows the calculation of the incorporation rates of biogenic silica in $\mu$mol BSi L-1 d-1units, as it is referred to the concentration of PDMPO incorporated in a specific volume of sample (250 mL) during one day of incubation. - To determine the silica incorporation rate we followed the standard procedure described in Leblanc and Hutchins (2005) and Shimizu et al. (2011). This measurement is an incorporation rate, a time-related parameter. We estimated the silica incorporation rates per unit of diatom biomass in the revised manuscript (values

shown in lines 5-6, page 8), and observed that this ratio did not show any significant relationship with increased temperature or pCO2. Silicification is performed by active cells, although the measurement of biomass is not related to the state of the cells and includes no actively growing cells and a component of detritical biomass. Probably, the presence of non active cell biomass influenced the incorporation rate vs. biomass ratio and prevented us for finding clear responses of the ratio with increased temperature or pCO2. The incorporation rates showed here reflecting the silicification process help us to identify the overall silicification responses of the communities and thus the consequences for the biogeochemical cycles.
* * *
Interactive
comment

Figure 1

[Figure]

**Fig. 1.** New Figure 1